# Epineural Neurorrhaphy of a Large Nerve Defect Due to IatroGenic Sciatic Nerve Injury in a Maltese Dog

**DOI:** 10.3390/vetsci9070361

**Published:** 2022-07-15

**Authors:** Hanjung Lee, Haebeom Lee, Keyyeon Lee, Yoonho Roh, Seongmok Jeong, Daehyun Kim, Jaemin Jeong

**Affiliations:** 1Department of Veterinary Surgery, College of Veterinary Medicine, Chungnam National University, 99, Daejeon 34134, Korea; aeonian10@gmail.com (H.L.); seatiger76@cnu.ac.kr (H.L.); kiyun948@hanmail.net (K.L.); jsmok@cnu.ac.kr (S.J.); vet1982@cnu.ac.kr (D.K.); 2Division of Small Animal Surgery, Department of Clinical Veterinary Medicine, Vetsuisse Faculty, University of Bren, 3014 Bern, Switzerland; yoonho.roh@vetsuisse.unibe.ch

**Keywords:** epineural neurorrhaphy, iatrogenic neurotmesis, traumatic sciatic neuropathy, femoral head and neck osteotomy complication, dog

## Abstract

**Simple Summary:**

Sciatic nerve injury could occur due to mistake of surgery and called as ‘iatrogenic injury’. This type of injury is rare in dogs. Historically, this injury is treated through physiotherapy. However, if the nerve is completely transected, surgery such as nerve repair could be addressed. Unfortunately, if there is a large gap between transected sciatic nerve, it is very difficult to treat. Sometimes amputation is recommended because of permanent problem with dog’s hind leg. By the way, it is not known how long the gap can be treated in dogs before the important decision of whether to amputate the leg or not. Therefore, we would like to described a good result of treating an iatrogenic sciatic nerve injury with a large defect measuring 20 mm in length in a small Maltese dog. The dog suffered nerve injury after hip joint surgery and could not be walking himself for 2 months. So, we decided to treat him by nerve repair despite of large gap. Sensation and walking function of his hind leg was recovered almost completely after 2.5 years. Although sciatic nerve injury with large gap is a concern, it could be treated through surgery, even in small Maltese.

**Abstract:**

Epineural neurorrhaphy is a standard nerve repair method, but it is rarely reported in veterinary literature. Epineural neurorrhaphy in canine sciatic nerve injury are described in this report. An 11-month-old, castrated male Maltese dog, presented with an one-month history of non-weight bearing lameness and knuckling of the right pelvic limb. The dog showed absence of superficial and deep pain perception on the dorsal and lateral surfaces below the stifle joint. The dog had undergone femoral head and neck osteotomy in the right pelvic limb one month prior to referral at a local hospital. Based on physical and neurological examinations, peripheral nerve injury of the right pelvic limb was suspected. Radiography showed irregular bony proliferation around the excised femoral neck. Magnetic resonance imaging revealed sciatic nerve injury with inconspicuous continuity at the greater trochanter level. A sciatic nerve neurotmesis was suspected and surgical repair was decided. During surgery, non-viable tissue of the sciatic nerve was debrided, and epineural neurorrhaphy was performed to bridge a large, 20-mm defect. The superficial and deep pain perception was progressively improved and restored at 3 weeks postoperatively, and the dog exhibited a gradual improvement in motor function. At 10 weeks postoperatively, the dog showed no neurological deficit including knuckling but the tarsal joint hyperextension did not improve due to ankylosis. The dog had undergone tarsal arthrodesis and exhibited almost normal limb function without any neurologic sequela until the last follow-up at 2.5 years postoperatively.

## 1. Introduction

Sciatic nerve injury usually develops secondary to a pelvic or femoral fracture, hip dysplasia [1,2] or iatrogenically during the repair of these fractures [3]. The proximal portion of the sciatic nerve is located deep near the bone and covered by deep and middle gluteal muscles, and the distal portion of the sciatic nerve is covered by biceps femoris, semitendinosus, and semimembranosus muscles. The distal portion of the sciatic nerve is located relatively shallow than the proximal portion. Therefore, the distal portion of the sciatic nerve is more susceptible to injury than the proximal portion [4]. The typical clinical features of sciatic nerve injury are claudication of the affected limb, mono-paresis with impairment of the lower motor neuron, proprioceptive deficits, and severe muscle atrophy.

Sciatic nerve injury is classified into three types based on Seddon classification: neuropraxia, axonotmesis, and neurotmesis [5,6]. Surgical treatment is indicated when there is complete disruption of the nerve, namely, neurotmesis [7]. The prognosis depends on several factors, such as the nature of the wound, date of suturing, patient-related factors, and surgeon-related factors [8,9]. Direct nerve neurorrhaphy which is the surgical suturing of both divided nerve ends is considered a standard treatment for nerve injury [10]. The techniques of direct nerve neurorrhaphy include standard epineural suture, fascicular suture, epineural sheath suture, single fascicle suture, and conduit suture [11]. Alternatively, other adhesive biomaterials including fibrin glue can be used for adhesion [11,12].

However, if there is a large gap between proximal and distal ends of the injured nerve and the nerve is anastomosed under high tension, pain or intraneural hemorrhage occurs, followed by scar formation and axoplasmic deterioration, which can result in poor functional recovery [13]. Moreover, such damage can cause nerve fiber contraction and inhibit nerve fiber maturation and proper myelinization [8]. In these cases, it is necessary to adopt alternative methods such as grafting, use of neuro-tubes [14], anchoring stitches [15], collagen nerve guide tube [16], or other methods that can reduce the tension [17].

These alternative methods are recommended when the nerve gap is 20 mm or more in human literature [18]; however, in veterinary medicine, there is no consensus regarding the size of the nerve gap up for which a direct neurorrhaphy can be performed. In this report, we describe a case of iatrogenic sciatic nerve injury with a 20-mm large gap in a Maltese dog that has been successfully treated with epineural neurorrhaphy.

## 2. Detailed Case Description

An 11-month-old, weighting 3.5 kg, castrated male Maltese dog was referred for non-weight-bearing lameness and knuckling of the right pelvic limb. One month prior to referral, the dog had undergone femoral head and neck osteotomy (FHNO) in the right pelvic limb at a local veterinary hospital due to Legg-Calve-Perthes Disease. The dog showed proprioceptive deficits and absence of superficial and deep pain perception of operated limb immediately after surgery. Thigh muscle mass was decreased progressively. There was no improvement in neurological signs despite of physical rehabilitation at our hospital, which included passive range of motion exercises, balance equipment exercises, electroacupuncture, neuromuscular electrical stimulation, and laser therapy.

In physical examination, the dog showed pain reaction during palpation of the right hip joint and the right tarsal joint was hyperextended and had decreased passive range of motion compared to the normal opposite limb. Previous skin incision was observed caudal to the greater trochanter. Neurological examination revealed a 20% decrease in right thigh muscle mass compared with the contralateral limb, mono-paresis, proprioceptive and postural reaction deficits of the right pelvic limb (Figure 1). The thigh muscle mass measured the circumference of the convex portion of the thick muscle mass, which is one-third the distal position of the length from greater trochanter to the femur epicondyle. In addition, withdrawal reflex was decreased, and cranial tibial reflex was absent in the right pelvic limb. However, the right patellar reflex was increased. The superficial pain perception was absent on the dorsal and lateral surfaces of the right pelvic limb below the stifle joint but was present on the plantar and medial surfaces. The deep pain perception was only elicited on 1st and 2nd metatarsal bone. Neuroanatomical localization was consistent with right sided sciatic nerve involving peroneal nerve branches.

Based on the patient’s clinical signs, history, and the results of physical and neurological examinations, iatrogenic right-sided sciatic neuropathy was suspected since the neurological deficit was shown immediately after the previous operation and the skin incision was caudal to the greater trochanter where the sciatic nerve was located. The patient was sedated with intramuscular administration of medetomidine (30 μg/kg). Radiographic examination including ventrodorsal hip extended and lateral view of the pelvis and craniocaudal and open-leg mediolateral view was performed for the right coxofemoral joint. Despite of previous FHNO surgery, irregular bony proliferations around the osteotomy line and acetabular rim were identified on radiographs. Magnetic resonance imaging performed under general anesthesia revealed inconspicuous continuity of the right sciatic nerve at the greater trochanter level (Figure 2). In the transverse and dorsal planes, hypointense structures appearing as osteophytes were observed in the right greater and lesser trochanters and acetabulum on T1 and T2-weighted (W) and contrast enhanced fat-suppressed proton density weight images (CE-FS-PDWI). On T1W and T2W images, peripheral muscles were observed to be non-uniform in intensity due to fibrotic changes. Volumes of the right adductor, gluteal, quadriceps, biceps femoris, semitendinosus, and semimembranosus muscles were decreased compared with the left side, and fatty infiltration was present. These findings were consistent with sciatic nerve injury and muscle atrophy.

Two months after the initial surgery, there was no improvement in neurological symptoms despite rehabilitation treatment. Thus, surgical treatment was decided after consulting with the owner. The dog was premedicated with intravenous (IV) administration of hydromorphone (0.05 mg/kg) and midazolam (0.2 mg/kg). Cefazolin sodium (22 mg/kg IV) was used as a prophylactic antibiotic before surgery and was injected additionally every 90 min during surgery. General anesthesia was induced with propofol (2–4 mg/kg IV) and maintained with inhaled isoflurane and oxygen. Analgesia was provided by constant rate infusion (CRI) of remifentanil (0.1–0.3 μg/kg/min). Greater trochanteric osteotomy was performed to secure the field of view. Proliferated bone on the femoral neck was removed using a high-speed burr. Grossly, the sciatic nerve was ruptured completely and entrapped within fibrotic tissues. The dog was diagnosed with sciatic nerve neurotmesis in combination with preoperative examinations. Each end of the ruptured sciatic nerve was debrided for relieving tension. The size of the defect was measured as 20 mm using a ruler intraoperatively. The nerve ends were mobilized and tension was evaluated during the apposition of both nerve ends. The epineurium was circumferentially exposed by carefully pushing back the mesoneurium with Dumont forceps, and the neurorrhaphy was performed with 7-0 polydioxanone suture (PDS II, Ethicon, NJ, USA) through the epineural suture pattern (Figure 3). While proceeding with the epineural suture, placing the stitches on other than epineurium was avoided as possible. Initially, two simple epineural sutures were placed on medial and lateral sides, 180 degrees from one another. These sutures were not tied until other sutures are placed. Two additional sutures were placed on the cranial and caudal surface respectively and then all sutures were tied. The greater trochanter was reattached using a tension-band wire, and the surgical wound was closed as usual. The debrided fibrotic tissues between both proximal and distal ends of the sciatic nerve with a small amount of remaining nerve tissue were submitted for histopathologic examination, and the composition of the debrided tissue was identified as multifocal axonal degeneration, hemorrhage, and suture granulomas (Figure 4). Cefazolin sodium (22 mg/kg IV twice daily) was prescribed for seven days after surgery. Postoperatively, remifentanil (0.1 µg/kg/min CRI) was administered during the first 12 h and subsequently replaced with meloxicam (0.2 mg/kg subcutaneously once daily on day 1, 0.1 mg/kg orally once daily on days 2–5). Gabapentin (15 mg/kg orally twice daily) was prescribed for alleviating neuropathic pain. A soft padded bandage was applied for 24 h to prevent postoperative edema. At five days postoperatively, passive range of motion exercises (PROM), toe touching and leash walking for 5 min twice daily, and laser therapy and acupuncture twice a week were started as rehabilitation until three weeks after surgery. At three weeks postoperatively, the superficial pain perception from the cranial part up to the proximal quarter below the tarsus was recovered, and the dog was able to ambulate with intermittent knuckling. The dog showed improved proprioceptive, postural reaction and withdrawal reflex. The dog discharged with client education for PROM and toe touching twice daily at home and laser treatment, underwater treadmill and acupuncture were performed twice a week for seven weeks. At seven weeks postoperatively, superficial pain perception was present in the cranial part of the distal tarsal region. At ten weeks postoperatively, the dog was able to ambulate without knuckling, and the superficial pain perception was present, but tarsal joint hyperextension did not improve due to ankylosis. Finally, right tarsal arthrodesis was performed using double 1.2 mm titanium locking plates (Arixvet, Jeil Medical Corp., Seoul, Korea) placed on the dorsal surface of the tarsal joint with rhBMP-2 (Novosis, CGBIO, Seongnam, Korea) three months after the surgery (Figure 5). One of these locking plates and multiple screws were destabilized 7 months after tarsal arthrodesis and bony union was observed without any complications until last follow up.

Gait analysis was conducted 23 months after the surgery with the pressure-sensitive walkway system (CanidGait^®^, Zebris Medical GmbH, Isny, Germany); equal force distribution in left and right pelvic limbs was observed. The gait analysis was performed twice on the same day, and force distribution analysis was performed to confirm the maximum force and average pressure distributed to each limb. The average force applied to both pelvic limbs in two trials was measured. On average, left and right pelvic limbs supported 59.5% and 58% of body weight, respectively. There was almost no bias in the center of weight. The circumference of the right pelvic limb muscle increased by 26% at 18 months postoperatively. Patellar and withdrawal reflexes were normal distally and proximally in affected pelvic limb. The dog was able to ambulate close to normal with the increased muscle mass and improvement in nerve function at 23 months postoperatively.

## 3. Discussion

Iatrogenic sciatic nerve injury is a rare condition in dogs [3]. In the present case, a large nerve gap of 20 mm caused by iatrogenic sciatic nerve injury in a small dog was repaired using epineural sutures, which resulted in good clinical recovery. The clinical features such as muscle atrophy and superficial pain perception improved significantly postoperatively.

The peripheral nerve injury is classified into three groups by Seddon and Sunderland [19]. Neurapraxia, a local myelin injury of neurons usually secondary to compression or traction without structural nerve damage. Axonotmesis, is the loss of continuity of axons. It mainly follows a stretch injury. Axonotmesis usually results from a more severe crush or contusion injury than that causing neurapraxia. The axons and their myelin sheaths are damaged, but endoneurium, perineurium, and epineurium remain intact. Neurotmesis, is complete transection or disruption of the entire nerve. The nerve and its sheaths are both damaged. It is the most severe nerve injury, with the affected nerve losing all sensory and motor functions [20]. In the present case, the magnetic resonance image revealed a focal discontinuity of the sciatic nerve and soft tissue edema, and the damaged nerve was identified intraoperatively, indicating neurotmesis. Additionally, in the final histopathological examination of the damaged sciatic nerve, a mixture of suture granulomas, caused by the residual suture material in the previous surgery, and axonal degeneration was observed. Without treatment, neurotmesis of a segment of a peripheral nerve results in poor recovery of motor and sensory functions of the affected limb and often leads to limb amputation [20,21]. Debridement of the damaged nerve was performed, and nerve segments were connected using epineural sutures.

The technical aspects to be considered during nerve repair includes the handling of nerve tissues, tension at the suture site, aseptic field, and foreign body response to suture materials [22]. Gentle handling of tissues facilitated by appropriate magnification is essential. Moreover, a complete absence of tension at the suture site is regarded as one of the important factors required for a successful nerve repair [23]. Common suture methods for recovering simple nerve laceration include epineural, interfascicular, and perineural techniques. In the present case, the epineural suture was used because the patient was too small, and it was technically simple and could lead to a well-matched alignment of the nerve stumps compared to other methods [22,23].

Generally, autologous nerve graft is recommended for nerve repair in cases with a nerve gap of more than 20 mm to prevent tension at the suture site [18]. However, there is a considerable loss of sensitivity at the resection site of the graft tissue. Moreover, the availability of autologous donor tissue is limited [24]. Therefore, if appropriate tension reduction is possible, the epineural neurorrhaphy is selected first for sciatic nerve repair [25]. Despite this general consensus, very few clinical case reports of sciatic nerve injury have been reported in veterinary medicine, except in experimental studies [4,26]. In addition, to the best of the author’s knowledge, there is a lack of clinical study reported about neurorrhaphy for the treatment of sciatic nerve neurotmesis. In the present case, we reported direct repair could be performed through epineural neurorrhaphy after blunt separation around the damaged sciatic nerve with large gap.

The sciatic nerve is a mixed nerve that derives from L6–S1 spinal segments and is responsible for vital motor and sensory functions in both limbs. These functions include movements of the posterior thigh and hamstring portion of the adductor magnus, which indirectly influence the movements of all appendicular muscles. The terminal branches of the sciatic nerve indirectly innervate the skin of the lateral pelvic limb, tarsal joint, and both the dorsal and plantar surfaces of the metatarsus to provide sensory function. The recovery time after epineural neurorrhaphy varies depending on the degree of nerve damage. Herein, the evaluation was conducted at 1–2-week intervals depending on the period described in previous studies [12,26,27]. It was confirmed that sensory nerve function was recovered three weeks after surgery, and motor function was recovered 10 weeks after surgery.

Gait analysis performed 23 months after surgery revealed that functions of the right pelvic limb were comparable with those of the healthy left pelvic limb. Force plate and pressure-sensitive walkway systems are widely used and validated for analyzing gait and evaluating limb function [28,29,30]. The pressure-sensitive walkway system has the advantage of being able to measure multiple gait cycles and ground reaction force of all limbs in one pass compared to the force plate [29]. Furthermore, temporospatial data such as velocity, acceleration, stance, and stride time could be able to obtain at once with ground reaction force. Especially in small dogs, it is difficult to obtain only 1 paw strike at a time using the standard force plat size. Therefore, pressure-sensitive walkways can be more versatile regarding variable patient sizes. We reported average maximum force expressed as % of body weight through an average of two trials. The symmetry difference of both the right and left pelvic limbs was found to be 1.5%. Previous studies have reported that the symmetry difference between the right and left limbs normally varies by 3.2% to 6% [31,32].

Sciatic nerve neuropathy impairs innervation of the distal branch including the peroneal and tibial nerves and can lead to the dropped hock joint and digital knuckling [4,33]. Depending on the severity, the pelvic limb could support weight through quadriceps muscles [4,34]. Additionally, rehabilitation and orthotics can be used to assist in walking in mild sciatic nerve neuropathy including transient neurapraxia. However, it has been reported that severe injuries such as neurotmesis have a poor prognosis without surgical treatment. In our case, the patient had a delay of two months after the initial sciatic nerve injury and showed severe pelvic limb dysfunction. Although the sciatic nerve was repaired through epineural neurorrhaphy, tarsal joint ankylosis and mild weight bearing lameness remained. However, the neurologic function was improved especially motor function, and muscle mass and weight bearing were increased after surgery. Consequently, the gait function was improved and the owner was satisfied with the results. Further, the dog restored the affected limb function close to normal through tarsal arthrodesis.

The limitation of this report is that the degree of axonal regeneration and muscle re-innervation were not evaluated histologically after recovery. In addition, tarsal joint ankylosis was not recovered despite of neurological restoration after surgery. Lastly, although our case showed satisfactory outcomes after epineural neurorrhaphy, we did not evaluate repaired nerve function through the pre and postoperative electro-diagnostics including electromyography and nerve conduction study. However, our case study described good clinical outcomes and may provide the clinical evidence that direct epineural neurorrhaphy could be performed for sciatic nerve neurotmesis with large defect of 20 mm in small breed dog.

## 4. Conclusions

The report suggests that in small dogs with sciatic nerve injury, end-to-end anastomosis can be considered if tension at the suture site is reduced through sufficient releasing despite a 20-mm gap between proximal and distal ends of the nerve. This case report also provides a strategy for managing iatrogenic sciatic nerve injury.

## Figures and Tables

**Figure 1 vetsci-09-00361-f001:**
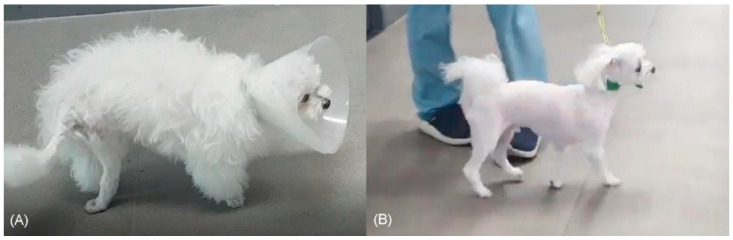
Preoperative and postoperative patient images. Posture observation before (**A**) and after (**B**) sciatic nerve neurorrhaphy and tarsal arthrodesis. The muscle atrophy of the right pelvic limb preoperatively is evident (**A**). At 21 months postoperatively (**B**), the muscle mass increased considerably, and the patient was able to bear weight on the operated limb.

**Figure 2 vetsci-09-00361-f002:**
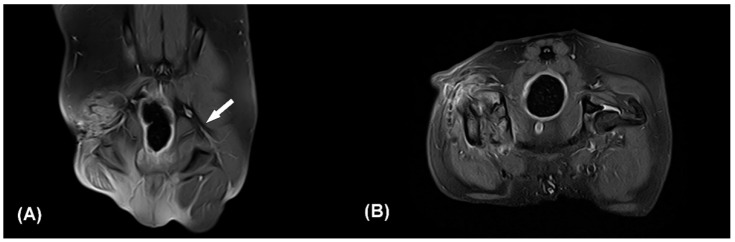
Contrast enhanced fat-suppressed proton density weight images (CE-FS-PDWI) of dorsal (**A**) and transverse plane (**B**). The white arrow indicates the intact left sciatic nerve (**A**). The right sciatic nerve strand is not clearly identifiable (**A**,**B**). Note that focal discontinuity of the sciatic nerve, muscle fibrosis, and edema of surrounding soft tissue. Q: quadriceps muscle, Bi: biceps femoris muscle.

**Figure 3 vetsci-09-00361-f003:**
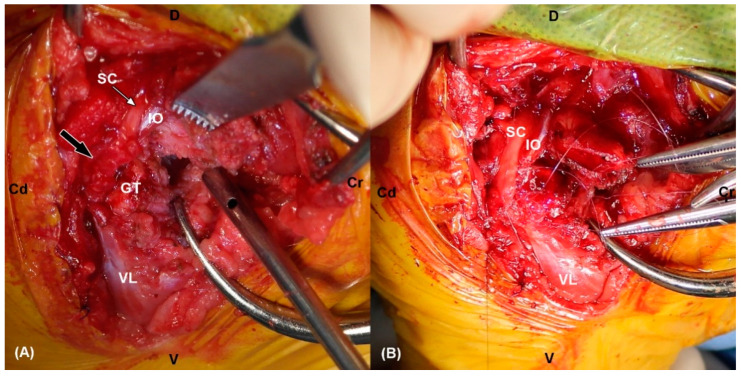
(**A**) The intraoperative photograph shows fibrotic (black arrow) and indistinguishable lesions of the right sciatic nerve. Note the intact proximal ending of sciatic nerve (white arrow). (**B**) Neurorrhaphy was performed by using multiple epineural sutures after debridement of the damaged nerve. SC: sciatic nerve, IO: internal obturator muscle, VL: vastus lateralis muscle, GT: greater trochanter, Cr: cranial, Cd: caudal, D: dorsal, V: ventral.

**Figure 4 vetsci-09-00361-f004:**
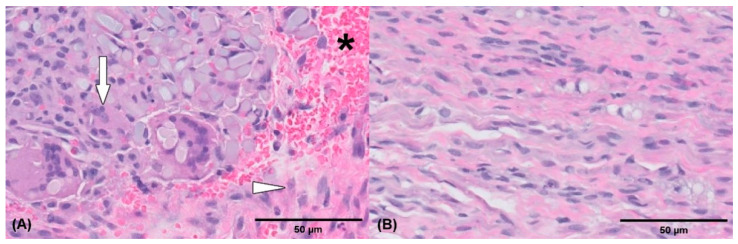
Histological pictures of the injured debrided sciatic nerve tissue (**A**,**B**). Extensive fibrosis (arrowhead) is present and partially surrounds the nerve. Multifocal axonal degeneration (arrow) and hemorrhage (asterisk) are also noted. Hematoxylin and eosin stain (actual magnification, 50× (**A**), actual magnification, 400×, (**B**)).

**Figure 5 vetsci-09-00361-f005:**
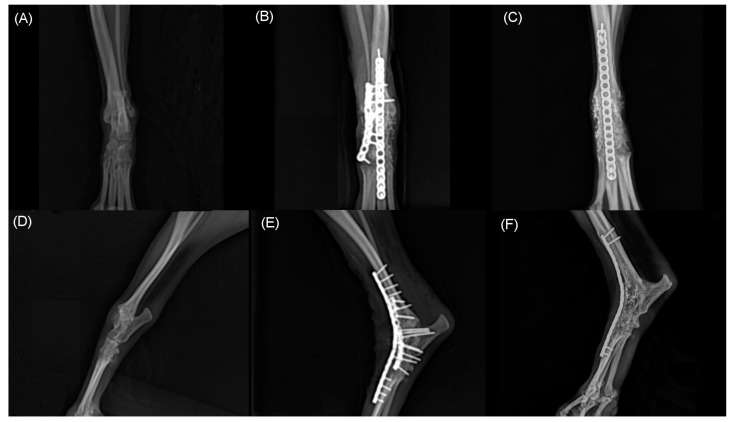
Craniocaudal (**A**,**B**,**C**) and mediolateral (**D**,**E**,**F**) radiographs of right tarsal joint. Note hyperextension of tarsal joint before surgery (**A**,**D**). Immediate postoperative radiographs (**B**,**E**) shows pantarsal arthrodesis performed using double 1.2 mm locking plate place on dorsally. At 21 months postoperative radiographs shows complete bony union and remodeling (**C**,**F**).

## Data Availability

Not applicable.

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
