# Peer review of "Epineural Neurorrhaphy of a Large Nerve Defect Due to IatroGenic Sciatic Nerve Injury in a Maltese Dog"

_vetsci, 2022, doi:10.3390/vetsci9070361_

Round 1
Reviewer 1 Report
Thank you for your submission. The manuscript is very well written and presented. There are areas of improvement regarding consistency, terminology and presentation of neurological examination. MRI pictures require editing. Also, further work and 'sciatic neurorraphy'-specific for dogs discussion should be attempted. Further and detailed comments and advice are provided in the pdf with track changes. At last, it is important to be highlighted why this study is novel compared to previously published in the literature. Thanks once again for your work.

Author Response
We thank the Reviewer for commenting in detail on important issues related to our study. We improve regarding consistency, terminology and presentation of neurological examination. Also, further discussion has been made as your suggestions.
Please see the attachment.

Reviewer 2 Report
Dear authors:
I like your contribution.
With my best regards.
Author Response
We sincerely appreciate your comments.
With our best regards.
Reviewer 3 Report
The article describes an interesting case in which sciatic nerve injury was well treated with a direct end-to-end anastomosis. The case description needs improvement, and the discussion requires a more thorough description of the gait analysis procedure.
I suggest minor revisions to improve the manuscript, after which it may be considered suitable for publication.
L 30-32: Please, provide a more detailed anatomical description
L 53: Human medicine
L 65: Please, describe the method to detect the thigh muscle mass
L 66-67: What do you mean for decreased ROM?
In my opinion, the voluntary movement of the hook was not possible but was not present as a joint disease that decreased the range of motion. Please rephrase to explain better.
L 79-80: Please, could you specify the projections?
L 80-82: FHNO was performed to treat which disease?
Were the detected osteophytes present before surgery?
Is osteophyte formation possible within one month of the surgery?
Please, rephrase to explain better.
L 103: decompressed: in my opinion, it is not correct. I guess that "debridement" is enough to describe the procedure.
L 104: The size of the defect was measured on MRI?
If you measured the gap in the surgical field, please describe how you did it.
L 107: add: as usual
L 108: Does this mean the involvement of the nerve in muscle reconstruction?
Please, clarify and rephrase the sentence.
L 125: How long were these procedures performed, and how often?
L 132: operation change in surgery
L 133-135: Please, define if you performed a force gait plate analysis or a paw pressure analysis.
It is important to understand the results better
L 145: Please add: 20mm
L 149: delete classic
L 150: remove first
L 155: remove third
L 158: change disrupted is damaged
L156-160: The sentence is not clear,
Briefly: clinically, the dog sounded lame after surgery, and a neurological deficit was detected. Then,
the MRI confirmed the diagnosis of Sciatic nerve injury.
The damaged nerve was visualised during the second surgery on the surgical field.
Please, if this description is correct, rephrase the sentences
L 200-203: The force plate gait analysis is the gold standard to assess the limb recovery function objectively. Please, provide a detailed description of the procedure employed and a deep discussion of the results in this section.
Author Response
Thank you for thoughtful review of our article with interest. We added a paragraph about gait analysis according to your comment. We also improved the minor revisions as you mentioned.
Please see the attachment.

Reviewer 4 Report
Dear Authors,
please find below my suggestions:
INTRODUCTION
Line 1-2: unfortunately sciatic nerve injury may also -be associated to corrective procedures performed in dogs affected by hip dysplasia.
Please add the following reference:
- Tavola et al VCOT 2021 doi: 10.1055/s-0041-1735288. Epub 2021 Sep 29.
- Properzi et al Veterinary Science 2022 https://doi.org/10.3390/vetsci9060259
Case description
Why did you not perform a EMG preop?
Ln 82 - you perform X-rays and MRI under general anesthesia I suppose? please specify
Ln 100 - Please specify how many days passed between the FHNO and your surgery.
Ln 104-106 Clarify give more details about the suture technique you performed.
Ln 119 Did you give cefazoline just postoperatively? Why?
Ln 119-130: please better explain the post period. In particular: how long the dog was hospitalized? When did you schedule the rechecks? When did you start physiotherapy? How long the physiotherapy was?
Ln 130-131 please give more details regarding the tarsal arthrodesis (implants, surgical approach recheck etc..) and X-Rays as well (at least post op)
Discussion
I suggest to extend your discussion to previous reports published in Vet Med (if present) and in human medicine as well
In addition I would like you discuss
- why you did not perform the EMG in the preop period, I think it could be useful to address the sciatic nerve function.
- The role of arthrodesis regarding the outcomes
- Potential complications associated to this nerve repair
- You successfully performed the surgery at least one month after the injury. Do we have a cutoff period after which the nerve repair is useless?
Please discuss your rationale to perform the revision surgery
Kind regards
Author Response
We would like to thank to the Reviewer for suggestions on important issues related to our study. We revised manuscript according to Reviewer’s suggestions.
Please see the attachment.

Round 2
Reviewer 4 Report
Dear Authors,
thank you for addressing my suggestions.